# Flexible Impedimetric Electronic Nose for High-Accurate Determination of Individual Volatile Organic Compounds by Tuning the Graphene Sensitive Properties

**Tianqi Lu** [1,†] , **Ammar Al-Hamry** [1,*,†] , **José Mauricio Rosolen** [2] , **Zheng Hu** [1] , **Junfeng Hao** [1] , **Yuchao Wang** [1] , **Anurag Adiraju** [1] , **Tengfei Yu** [1] , **Elaine Yoshiko Matsubara** [2] and **Olfa Kanoun** [1,*]

[1] Chair Measurement and Sensor Technology, Department of Electrical Engineering and Information Technology, Chemnitz University of Technology, 09107 Chemnitz, Germany; tianqi.lu@etit.tu-chemnitz.de (T.L.); Zheng.Hu@etit.tu-chemnitz.de (Z.H.); Junfeng.Hao@s2017.tu-chemnitz.de (J.H.); Yuchao.Wang@s2017.tu-chemnitz.de (Y.W.); Adiraju.Anurag@etit.tu-chemnitz.de (A.A.); Tengfei.Yu@s2017.tu-chemnitz.de (T.Y.)

[2] Departamento de Química, Faculdade de Filosofia, Ciências e Letras de Ribeirão Preto, Universidade de São Paulo, Avenida Bandeirantes 3900, Ribeirão Preto 14040-901, SP, Brazil; Rosolen@ffclrp.usp.br (J.M.R.); Elainematsubara@yahoo.com (E.Y.M.)

[*] Correspondence: Ammar.Al-Hamry@etit.tu-chemnitz.de (A.A.-H.); Olfa.Kanoun@etit.tu-chemnitz.de (O.K.)

[†] Co-first author, these authors contributed equally to this work.

**Abstract:** We investigated functionalized graphene materials to create highly sensitive sensors for volatile organic compounds (VOCs) such as formaldehyde, methanol, ethanol, acetone, and isopropanol. First, we prepared VOC-sensitive films consisting of mechanically exfoliated graphene (eG) and chemical graphene oxide (GO), which have different concentrations of structural defects. We deposited the films on silver interdigitated electrodes on Kapton substrate and submitted them to thermal treatment. Next, we measured the sensitive properties of the resulting sensors towards specific VOCs by impedance spectroscopy. We obtained the eG- and GO-based electronic nose composed of two eG films- and four GO film-based sensors with variable sensitivity to individual VOCs. The smallest relative change in impedance was 5% for the sensor based on eG film annealed at 180 °C toward 10 ppm formaldehyde, whereas the highest relative change was 257% for the sensor based on two-layers deposited GO film annealed at 200 °C toward 80 ppm ethanol. At 10 ppm VOC, the GO film-based sensors were sensitive enough to distinguish between individual VOCs, which implied excellent selectivity, as confirmed by Principle Component Analysis (PCA). According to a PCA-Support Vector Machine-based signal processing method, the electronic nose provided identification accuracy of 100% for individual VOCs. The proposed electronic nose can be used to detect multiple VOCs selectively because each sensor is sensitive to VOCs and has significant cross-selectivity to others.

**Keywords:** graphene; reduced graphene oxide; electronic nose; VOC sensor; sensor array; impedance spectroscopy; PCA; SVM

## 1. Introduction

Measuring volatile organic compounds (VOCs) has become important for applications ranging from daily indoor air quality monitoring [1] to the clinical diagnoses of diseases (e.g., respiratory diseases [2–6] and gastrointestinal diseases [7]) and body fluids analysis [8]. Gaseous chromatography-mass spectrometry (GC-MS) is a high-performance measurement method to measure VOCs. However, it requires expensive equipment, is difficult to operate, and provides complex results to analyze [9]. In this scenario, commercial miniaturized and integrated devices are being developed rapidly, but portable instruments still cannot reach the same level of accuracy as professional equipment.

Portable biomedical devices should be sensitive to human health-related VOCs such as acetone and formaldehyde, but not as mature and stable as big equipment [10]. Given the relevance of sensors and array signal analysis methods, portable devices can be developed based on a multi-sensor array; that is, an electronic nose (e-nose, EN) [11], so that measured gas concentrations are not based on the response of an individual sensor. EN uses the entire sensor array response with signal processing to identify the so-called "fingerprint" of the gas [12].

Many sensitive materials for detecting VOCs can be prepared by the optimization of material composites and their preparation parameters such as chemical and thermal treatment, functionalization, and doping [13]. Recent studies have pointed out that graphene-based nanocomposite sensors can be well integrated into an EN, helping to detect relative humidity (RH), carbon oxides, and VOCs [14–16]. Well-established techniques can be used to prepare graphene-based materials, including mechanical cleavage [17], chemical vapor deposition [18], and reduction of graphite oxide [19]. Meanwhile, sensitive films can be prepared by methods such as drop-casting [20], spin coating [21], and Layer-by-layer (LBL) [22]. The potential of graphene and its derivatives in fabricating high-performance sensors was demonstrated in [13]. Oxygen gas can be measured at a ppt level by chemoresistive sensors based on chemical vapor deposition (CVD) of graphene [23]. Oxygen-containing functional groups such as hydroxyl and carboxyl groups on reduced graphene oxide (rGO) can work as adsorption sites for interaction with vapor molecules of $NH_3$ [24], causing changes in electrical properties. Lipatov et al. [25] drop-casted GO films on platinum interdigitated electrodes and thermally reduced them, to distinguish various VOCs. It was proven that the graphene-like layers have excellent sensitivity toward alcohols such as ethanol and n-butanol [26,27]. Functionalized graphene materials have been validated for detecting gases and can be employed to design highly selective sensors for certain gases. A silver nanoparticle-naphthalene-1-sulphonic acid reduced graphene oxide composite (Ag-NA-rGO) sensor has low limit of detection (LOD) (1 ppm) for $NO_2$ and excellent reproducibility [28]. $GO-SnO_2$ composite film sensors prepared by electrophoretic deposition (EPD) detected formaldehyde at ppm level [29]. Graphene and poly(methyl methacrylate) composite showed an acceptable response towards dichloromethane [30] and tetrahydrofuran [31]. Therefore, an EN composed of graphene materials might be a solution to recognize VOCs. Liu et al. [32] designed a sensor array based on GO modified by diverse amine ligands, to recognize 25 ppm of ethanol, 2-ethylhexanol, nonanal and ethylbenzene by Principal Component Analysis (PCA). Xu et al. [33] developed a wearable EN based on four porphyrin-modified rGO sensors for physiological signal monitoring and showed that the EN can identify eight VOC biomarkers. Nag et al. [34] employed the supramolecular assembly technique to fabricate a functionalized cyclodextrins/graphene composite sensor arrays to detect eight common VOCs. Nanocomposite sensors that work under direct current (DC) have sensitivity down to the ppb level, but response and recovery speed are poor, usually in the range of 10 to 100 s [35]. However, selectivity and response/recovery time can be improved when an alternating current (AC) impedance measurement technique is used [36,37]. Impedance measurement enhances the surface state dynamics and sensing response at high frequency. Kulkarni et al. [23] reported that graphene could achieve rapid response (less than 0.1 s) and a low limit of detection (ppb level) towards various VOCs including cis- and trans-isomers.

This article discusses the fabrication, sensing properties, and signal analysis of a novel graphene-based impedimetric EN to identify and to quantify VOCs associated with human health. More specifically, we use scalable methods to prepare graphene samples with different structural defects and a simple casting method to deposit the sensitive films. We investigate how the thermal treatment temperatures and the number of coating layers affect the response of the sensors to VOCs and analyze the calibration curves obtained for the mechanically exfoliated graphene (eG) and GO films towards formaldehyde, methanol, acetone, ethanol, and isopropanol, as recognizable VOCs. Impedance measurement was used to characterize VOCs' relative change and selectivity and determine the optimal

operating frequency range. Repeated measurements have confirmed the long-term stability of the sensors. PCA has shown that obtaining an EN for the highly accurate and quantitative recognition of VOCs is feasible. The accuracy of the PCA—support vector machine (SVM) method towards various VOCs has also been investigated and discussed. The proposed EN prepared with two eG and four reduced GO could be applied in clinical diagnosis of respiratory diseases and indoor air quality monitoring, particularly in the precise detection and quantitative analysis of harmful gases and biomarkers at concentrations as low as 10 ppm.

## 2. Materials and Methods

### 2.1. Material and Sensor Preparation

The proposed sensor array consists of graphene-based films (see Figure 1a). Two types of graphene materials were used, the first was exfoliated graphene (eG) which was prepared by the exfoliation method using graphite flake. The dispersion was in sodium dodecyl sulfate (SDS), and its zeta potential and size were approximately $-46$ mV and 285 nm, respectively. The other type was graphene oxide (GO) which was prepared using the Hummers method, with a concentration of 1 g/L GO in deionized water. The graphite was Brazilian natural graphite with flakes of big grain dimension.

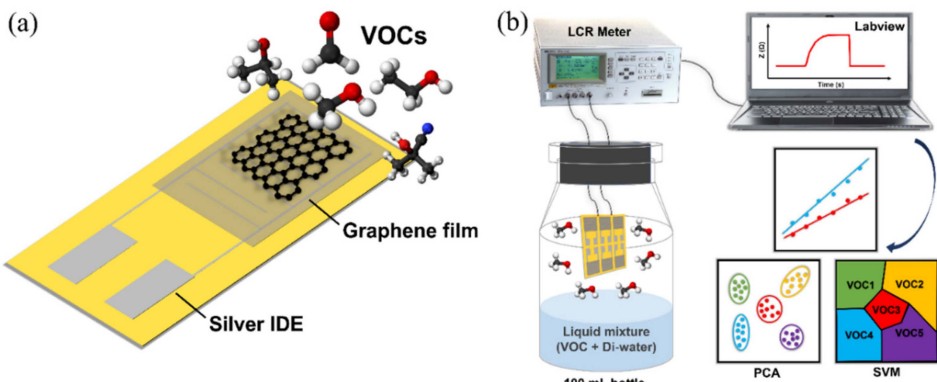

**Figure 1.** (**a**) Schematic representation of the proposed design of the graphene-based silver IDE sensor under target VOCs and (**b**) system overview of the impedimetric measurement setups and the corresponding signal processing.

Kapton HN plastic substrates were used; they were was initially cleaned in an ultrasonic bath followed by rinsing in deionized water several times and dried under a nitrogen ($N_2$) flow. Then the electrical contact of the sensor was designed as interdigitated electrodes (IDE) using inkjet-printed silver on the substrate, four pairs of electrodes (180 μm wide and 2.5 mm long per element) with a spacing of 330 μm.

For the preparation of the sensitive layer of the sensors, eG or GO dispersion was drop-casted on top of the silver IDE, limiting the film area using a mask prepared with pre-printed standard adhesive paper. The dropped volume was controlled at 10 μL. The sensors were made either as single-coated films or double-coated films, depending on the number of depositions. After deposition, the material was exposed to a dehumidified pure nitrogen environment to dry at room temperature.

The investigated sensor array comprised six eG- and GO-based sensitive films, and they were heated at 180, 200, and 250 °C for one hour in an oven under an air atmosphere before the VOC exposure testing.

### 2.2. Physical Characterization

The eG and GO particles used in the preparation of the sensors were characterized using atomic force microscopy (AFM) and Raman spectroscopy. For this purpose, they were deposited on a freshly cut silicon wafer by spin coating (4000 rpm, 40 s). The samples were analyzed by AFM (5600LS Atomic Force Microscope System, Keysight Technologies,

Santa Rosa, CA, USA) in tapping mode. In addition, 10 μL of GO was drop-coated on two silicon wafers. One of them was heated at 200 °C in the oven for one hour to compare with the unheated one. The same AFM device was used with setting in the CS-AFM mode to characterize the surface conductivity of the samples coated by drop-casting (sample bias 50 mV).

Three sets of samples were prepared for UV-vis characterization (Cary 60 UV-vis, Agilent Technologies, Santa Clara, CA, USA). Set 1 are the solution samples of 60 μL of eG (in SDS) and GO dispersions which were separately diluted in deionized water. Set 2 and 3 are eG and GO thin films which consisted of 10 μL of eG or GO dispersions that were individually coated on four transparent glass pieces by spin coating (2000 rpm, 20 s). To compare with the unheated samples, three of them were heated at 180, 200, and 250 °C.

Raman spectra were obtained by the Horiba Xplora Nano Instrument (Kyoto, Japan) at a wavelength of 532 nm; the acquisition time was two minutes. For Raman measurements, samples were prepared by drop-casting 10 μL of the GO- and eG-based solution on a silicon substrate and drying at room temperature (as for the investigated sensors preparation). The samples were also measured after applying thermal heating at 200 °C to verify the change by thermal annealing.

### 2.3. Gas Atmosphere Preparation

This work focused on the VOCs formaldehyde, methanol, acetone, ethanol, and isopropanol, commonly related to human health. If the air-water partition of the VOCs is considered [38–40], the concentration of VOC in the headspace air of bottle is controllable after 25 mL of water and a certain volume of VOC are added in a 100-mL sealed vial. In the experiment, the concentration in the bottle was controlled at five differentially increasing concentrations (10, 20, 35, 55, and 80 ppm), and the sensor array was placed in each bottle, in turn, until the response became stable. The initial impedance value is defined as the steady-state impedance response of a sensor in a bottle containing only deionized water (1 atm, 25 °C) to eliminate the effect of humidity on the sensor. The concentration of the headspace, $C_H$, is calculated as given in Equation (1) [41]:

$$C_H = \frac{C_w}{V_g/V_w + K_{aw}} \tag{1}$$

where $C_w$ is the concentration of the solution; $V_g$ and $V_w$ are the volume of the headspace and the solution, respectively; and $K_{aw}$ is the air-water partition of VOCs based on Henry's Law.

### 2.4. Impedimetric Electronic Nose

The EN was constructed from 6 sensors consisting of selected eG or GO, fabricated with tuned parameters (annealing temperature, amount of coating layers). Table 1 shows the fabrication details.

**Table 1.** Fabrication details of the six sensors constituting the proposed electronic nose.

| No. | Materials | Annealing Temp. (°C) | Number of Coating Layers |
|:---:|:---:|:---:|:---:|
| 1 | Exfoliated Graphene | 180 | 1 |
| 2 | Exfoliated Graphene | 180 | 2 |
| 3 | GO (rGO) | 200 | 1 |
| 4 | GO (rGO) | 250 | 1 |
| 5 | GO (rGO) | 180 | 2 |
| 6 | GO (rGO) | 200 | 2 |

The measurement of the electrochemical impedance spectroscopy of the array response towards VOCs was performed by the Agilent 4294A Precision Impedance Analyzer, as

shown in Figure 1b. The Nyquist and Bode plots of the range (40 Hz to 10 MHz) were analyzed to determine the optimal working frequency. The impedance measurement toward VOCs at a working frequency of 100 kHz were recorded by HP 4284A Precision LCR Meter for repeat measurements. Individual measurements were carried out thirty times for training and testing the machine learning.

### 2.5. Data Processing and Pattern Recognition

The response of the sensors toward VOCs is quantified as the relative change of impedance; i.e., the percentage ratio (%) between the impedance change $|\Delta Z|$ and the initial impedance $|Z_0|$, presented in Equation (2):

$$\text{Relative impendance change} = \frac{|\Delta Z|}{|Z_0|} \times 100\% \tag{2}$$

where the initial value $|Z_0|$ is defined as the stable absolute impedance value of sensors measured under the above-described measurement conditions without VOCs. The impedance change $|\Delta Z|$ is defined as the absolute difference between the measured impedance value Z and the initial value of sensors under the same measurement conditions with a certain VOC concentration.

The relative change of impedance was taken as input for the machine learning algorithms. PCA was applied for feature extraction, and SVM with a linear kernel (Linear-SVM) was used as a classifier. Ten sets of data were used for training, and twenty sets of data were used for testing the accuracy of the algorithm.

## 3. Results and Discussion

### 3.1. Physical Characterization

The structural eG and GO quality present in the respective aqueous dispersions used in the preparation of sensors and the possible morphology of GO and eG thin films obtained were probed using AFM characterizations. In Figure 2a, the eG films spin-coated on a silicon substrate were measured by tapping AFM modes. The continuous coating of the film can be observed, and single flakes can be identified. The height examination suggests having multiple-layer nanoplatelet graphene, where the average thickness was estimated to be ~4 ± 0.5 nm. In addition, some smaller platelets with multilayers of carbon are also detectable from the image with rather higher thickness.

In Figure 2b, the image of spin-coated GO on the silicon substrate reveals both well-dispersed solutions where individual flakes can be observed with sizes approaching one atomic layer thickness. Compared to eG flakes, GO flakes are larger as expected because the mechanical exfoliation of graphite flake is a process that can decrease the D50 depending on the sequence of millings used in its preparation. The statistics of the counts show that more than 50% of eG flakes have a size under 0.5 μm$^2$ (see Figure 2c). Figure 2d shows that most GO flakes have a size of approximately 0.2 μm$^2$, but smaller thickness than eG (~2 ± 0.3 nm). However, in some areas, some agglomerates are observed at the edges of the image due to the centrifugation applied on the materials on the substrate. Figure 2e,f are images of GO deposited by drop-casting on ITO glass where the topography and current sensing AFM (CS-AFM) characterization were carried out. The drop-coating of GO sheet produces thin films with visible wrinkles. The edge interactions are generally responsible for this kind of deformation in multi-layers of graphene or graphene oxide, mainly when the sheets are not in contact with the substrate, as evident in Figure 2a,b [42]. For drop-casted samples, aggregation is more apparent due to the larger film thickness and dispersion concentration of 0.1 wt.%. Notably, the current, from the CS-AFM image in the inset of each figure, increases after reduction at 200 °C. Whenever the thermal treatment was applied, the oxygen concentration in the GO thin films was expected to decrease, increasing the electronic conductivity of GO, which can be considered a reduced GO.

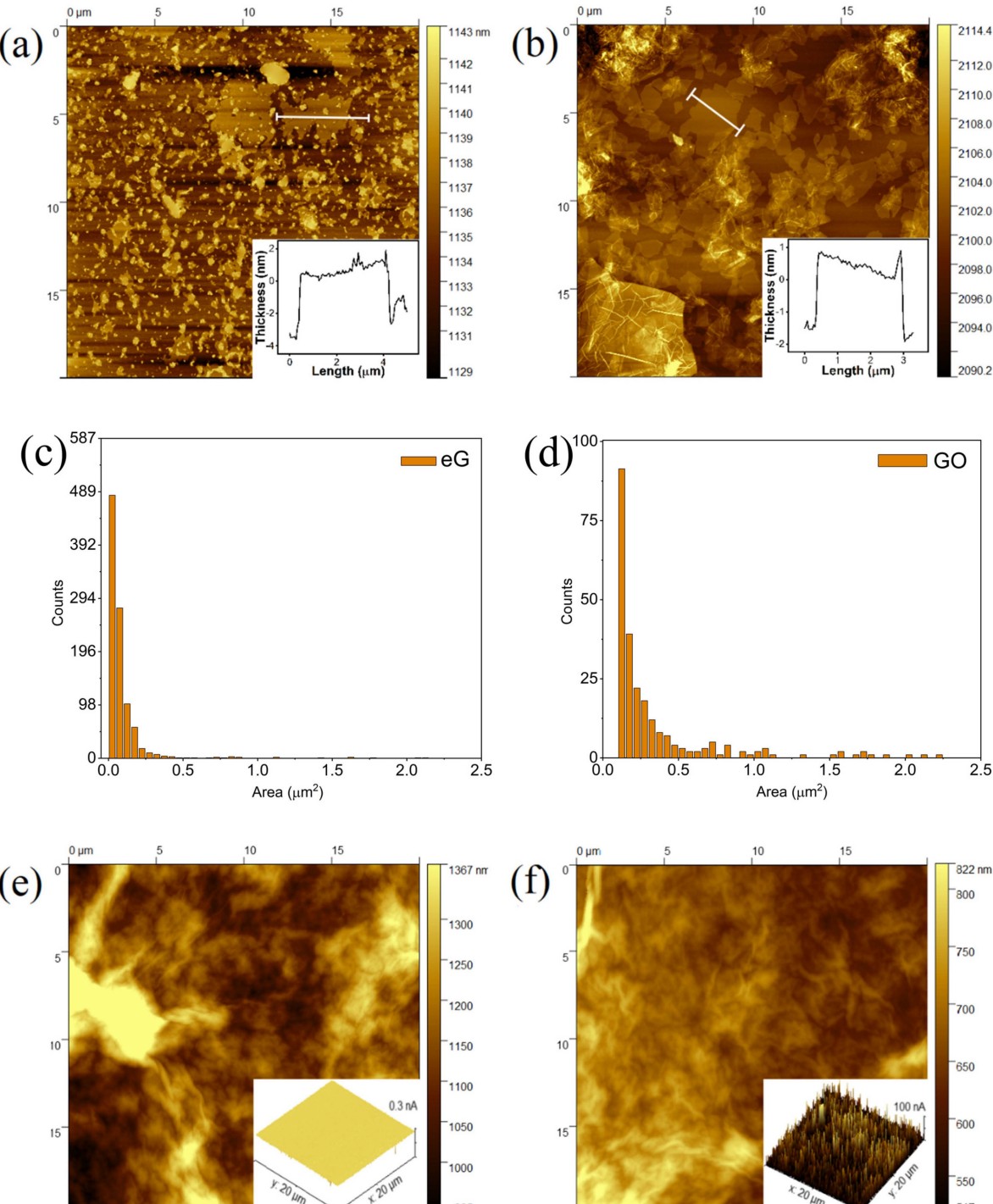

**Figure 2.** AFM and CS-AFM results for samples of (**a**,**b**) spin-coated eG and GO, respectively, and insets are the dimension of flakes; (**c**,**d**) the size statistic of spin-coated eG and GO from (**a**,**b**), respectively; for (**e**,**f**) drop coated GO before and after thermal annealing at 200 °C, respectively, and insets are the CS-AFM images.

Several measurements of UV-vis-NIR were also carried out to observe both aqueous dispersions and coated films on a glass substrate. In Figure 3a, GO/water and eG/SDS dispersions were tested after 50× dilution of each material in its solution, in quartz cuvettes. GO spectra show that the peak at 230 nm belongs to a π-π* transition of the C-C bond. The shoulder at 320 nm indicates the n-π* plasmonic transition of the C=O bonds [43]. Both confirm that the GO used in the present study contains only some graphene layers. While for eG, peaks at 230 and 265 nm indicate the presence of graphene bonds alone.

The correlations among the number of layers found in GO, rGO probed by AFM and the UV-vis spectra have been studied in the literature. Generally, GO sheets with layers below ten graphene layers present well-defined peaks in the UV-vis spectra, as observed in our samples [17]. In Figure 3b, it is possible to see the absorption peaks at 265 nm eG films are quenched due to the electronic interactions between graphene layers and substrate. However, the transmittance of single-layer CVD graphene is expected to be approximately 76% at 550 nm (~11 layers) and ~97% to CVD monolayer [44]. Hence, we suppose that graphene sheets used in the sensor preparation contain less than 10 layers because the transmittance of eG thin films is up to 86%. Figure 3c shows the spectra of GO and rGO films reduced at different reduction temperatures. For GO, both peaks and shoulders are observed apparently, such as in the case of respective dispersion. For reduced films, the shoulder disappeared, indicating the effective reduction already at these relatively low temperatures. In addition, the redshift of the peak from 230 to 265 nm is notably observed as a typical sign for rGO. The absorbance values generally increased for rGO films as they become darker than the brownish color of GO. The difference in absorbance values is probably due to thickness change compared to the original coated thicknesses.

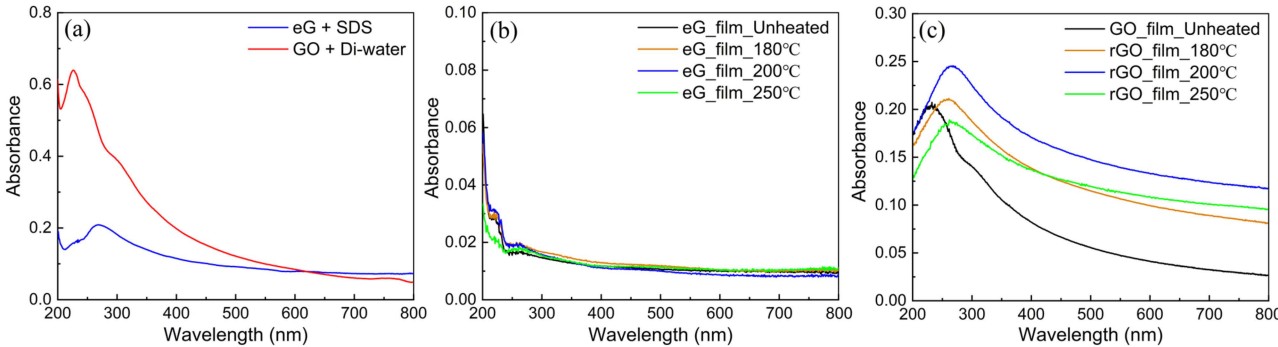

**Figure 3.** UV-vis-NIR results for (**a**) solution of eG and GO dispersions; and for (**b**) eG- and (**c**) GO-based films on optical glass.

Raman spectra are shown in Figure 4a,b for eG before and after heating, where typical graphene spectra are obtained with D and G peaks at 1341 cm$^{-1}$ and 1574 cm$^{-1}$, respectively, and a shoulder at 1600 cm$^{-1}$. Two peaks at ~2480 and 2900 cm$^{-1}$ are observed near the overtone of D peak ~2680 cm$^{-1}$ (2D band). Peaks between ~2500 to 2900 cm$^{-1}$ can contain several interband transitions as D + D″, 2D, and D + D′, while at approximately 1580 and 1620 cm$^{-1}$ we have the G and D′ transitions. Their presence will depend on graphene layers type (e.g., stacking and defects). The Raman spectra of nano graphite samples have D, 2D broaden peaks and further, the $I_D/I_G$ ratio increases and varies inversely with the length ($L_a$) of crystal size. The $I_D/I_G$ ratio also depends on the number of layers and defect density in the graphene layers [45,46]. Moreover, the D′ peak can also be observed near the G band, and D + D′ arises as well. Furthermore, the peak at 1620 cm$^{-1}$ (D′ peak) is attributed to the intravalley process occurring in graphene. It is a connection for two different points on the same circle [47]. Contrary to GO, graphene layers have an intense 2D peak, which decreases and shifts with an increasing number of graphene layers. The 2D band in the single graphene layer is the overtone of the D band, and it is associated with breathing modes of six carbon atoms and needs a defect to be activated. Finally, the intensity of the G band is also sensitive to doping effects or the presence of species such as oxygen groups or other molecules which can exchange charge with graphene layers [48].

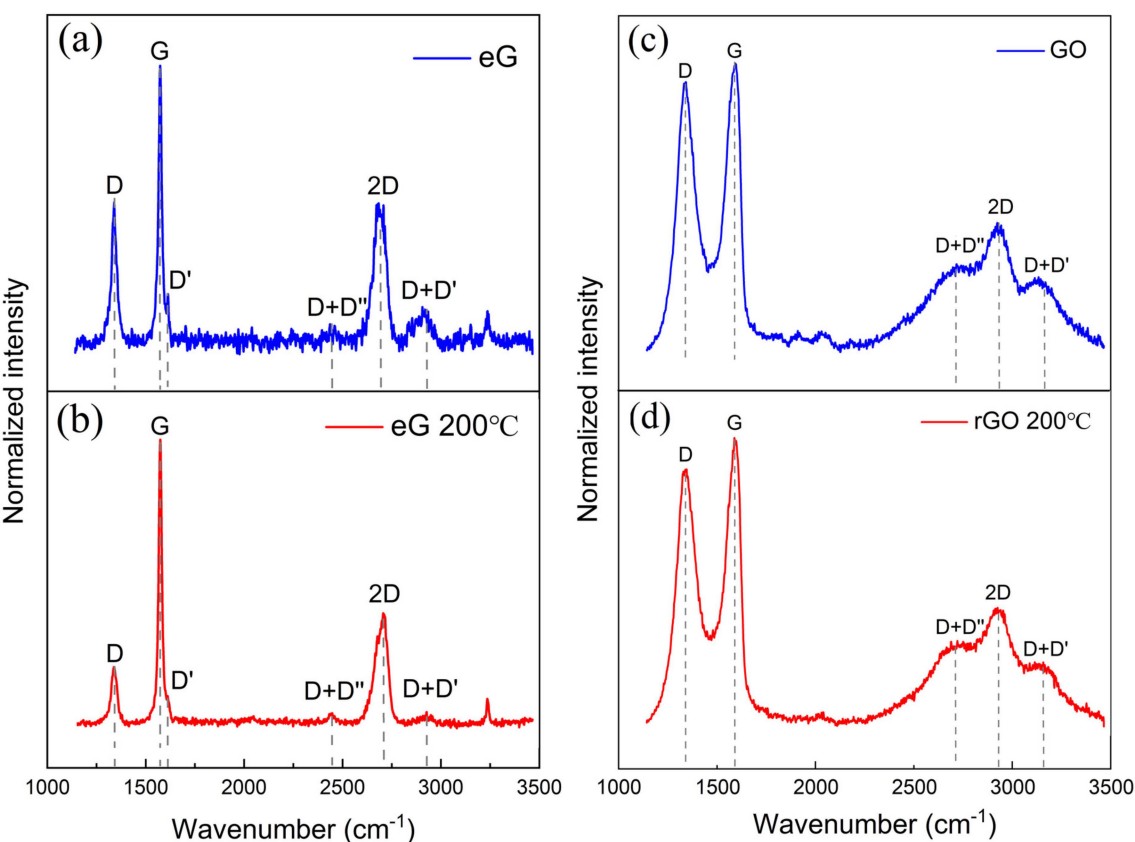

**Figure 4.** Raman spectra of eG (**a**) before and (**b**) after thermal annealing at 200 °C and of GO (**c**) before and (**d**) after thermal annealing at 200 °C.

Therefore, the eG Raman spectra corroborate the UV-vis spectra and AFM results. The resulting eG thin film prepared by casting is composed of a multilayer graphene and non-nano graphite. Additionally, we observed that the $I_D/I_G$ ratio reduced from 0.547 (Figure 4a) for eG to 0.2241 in the eG layer submitted to 180 °C (Figure 4b). This is predominantly due to the elimination of SDS from the surface of graphene, which would affect the intensity of the G band where SDS is a chemical species that can interact electronically with the graphene layer. The values obtained for heated eG lay in a similar range as reported in [49,50].

In Figure 4c,d, GO and rGO reduced at 200 °C are compared where D and G, D + D″, 2D, and D + D′ peaks are found at 1350, 1580, 2700, ~2940, and ~3170 cm$^{-1}$, respectively [51]. The chemical oxidation of graphene introduces defects in the graphene layers, which affects the intensity and the shift of the D bands. In addition, the oxygen groups in the edges and basal surfaces of graphene layers can also affect the intensity and position of the G peak.

By calculating the ratio of $I_D/I_G$ ratio it was found to be 0.93. After heating, the value changed to 0.89. Additionally, all D peaks were changed after the thermal treatment. These alterations reflect the decrease in the oxygen concentration groups, which decreased with heating. However, not all functional oxygen groups would be extracted, which would be interesting for VOCs detection, as discussed before. Typically, the -OH group is mainly removed from the GO lattice, and only with high reduction temperature treatment under vacuum is possible to observe 2D features similar to those of graphene [52].

### 3.2. Electrical Characterization

Electrochemical impedance spectroscopy is commonly applied to interpret the electrical characterization of sensors, and it can enable the selection of the optimal operating frequency specific to analytes [53,54]. For dynamic impedance measurements aiming at applications in VOCs detection, the operating frequency must be chosen and optimized

at a fixed value, so that all the concentrations and analytes can be measured. In this case, the Bode-plot can also help to select the operating frequency [37]. Equation (3) has been proposed to quantify the response amplitude at different frequencies on the basis of a cumulative calculation of the differences of response impedance:

$$\Delta Z_{Cumulative}(f) = \sum_{k=1}^{n} (||Z_k(f)| - |Z_0(f)||) \ (n = 10, \ 35, \ \cdots \text{ppm}) \tag{3}$$

where, $\Delta Z_{Cumulative}(f)$ is the frequency dependent cumulative impedance change, $|Z_0(f)|$ is the impedance value toward 0 ppm, and $|Z_k(f)|$ is the impedance value at the $k_{th}$ measured concentration.

   The Bode-plot of the sensor (eG, 180 °C, 1 L) toward different formaldehyde concentrations at different frequencies is shown in Figure 5a. Starting from 100 kHz, the multiple curves representing the concentration gradually converge, until finally indistinguishable, i.e., the cumulative value decreases significantly to a relatively low level, which is not suitable for pattern recognition. Therefore, any frequency below 100 kHz can be considered for this system, while a frequency above 100 kHz causes the sensor to be insensitive. To ensure the feasibility of the proposed system and algorithm in lower frequencies, 100 kHz is selected as the operating frequency in the experiments.

   Figure 5c,d respectively show the Nyquist plots for the sensor (eG, 180 °C, 1 L) for different formaldehyde concentrations and different types of VOCs at 10 ppm, at room temperature. A large and slightly depressed semicircular arc with a long tail can be observed in the examined frequency range (40 Hz to 10 MHz). In Figure 5c, the semicircle shows a unidirectional and regular shift toward low frequency with increasing formaldehyde concentration. The shift magnitude correlates with the response intensity. Thus, the sensor displays a monotonic and stable response to formaldehyde. the Figure 5d shows that the semicircles of five VOCs are also distributed differently, indicating that the sensor has promising selectivity, implying sufficient response to reducing VOCs at low concentrations. This phenomenon demonstrates the potential of mechanical graphene-based materials to recognize VOCs qualitatively and quantitatively under impedimetric measurement conditions, which can positively impact pattern recognition by the EN system.

### 3.3. Response on VOCs

   Figure 6a provides the relative change of impedance and the second-order polynomial fitting curve of the sensor (rGO, 200 °C, 2 L) toward five VOCs at room temperature. The response trends for different VOCs show excellent behavior and repeatability. They could even be fitted by the same second-order polynomial equation, with corresponding coefficients for each VOC. Failure in qualitative recognition due to intersection of response curves is avoided. The sensor has sufficient relative change toward different VOC concentration levels. The maximum achieved relative change is 257.9% when the ethanol at 80 ppm is added to the headspace. The relative change for toxic contaminants, such as formaldehyde and methanol, can also achieve 140.2% and 111.8% at 80 ppm, respectively. Figure S1 shows the dynamic process of the response towards methanol (sensor rGO, 200 °C, 2 L). The relative impedance change increases significantly by changing the concentrations of the headspace (0–80 ppm) and can recover well to its initial value in every case when putting back in 0 ppm headspace.

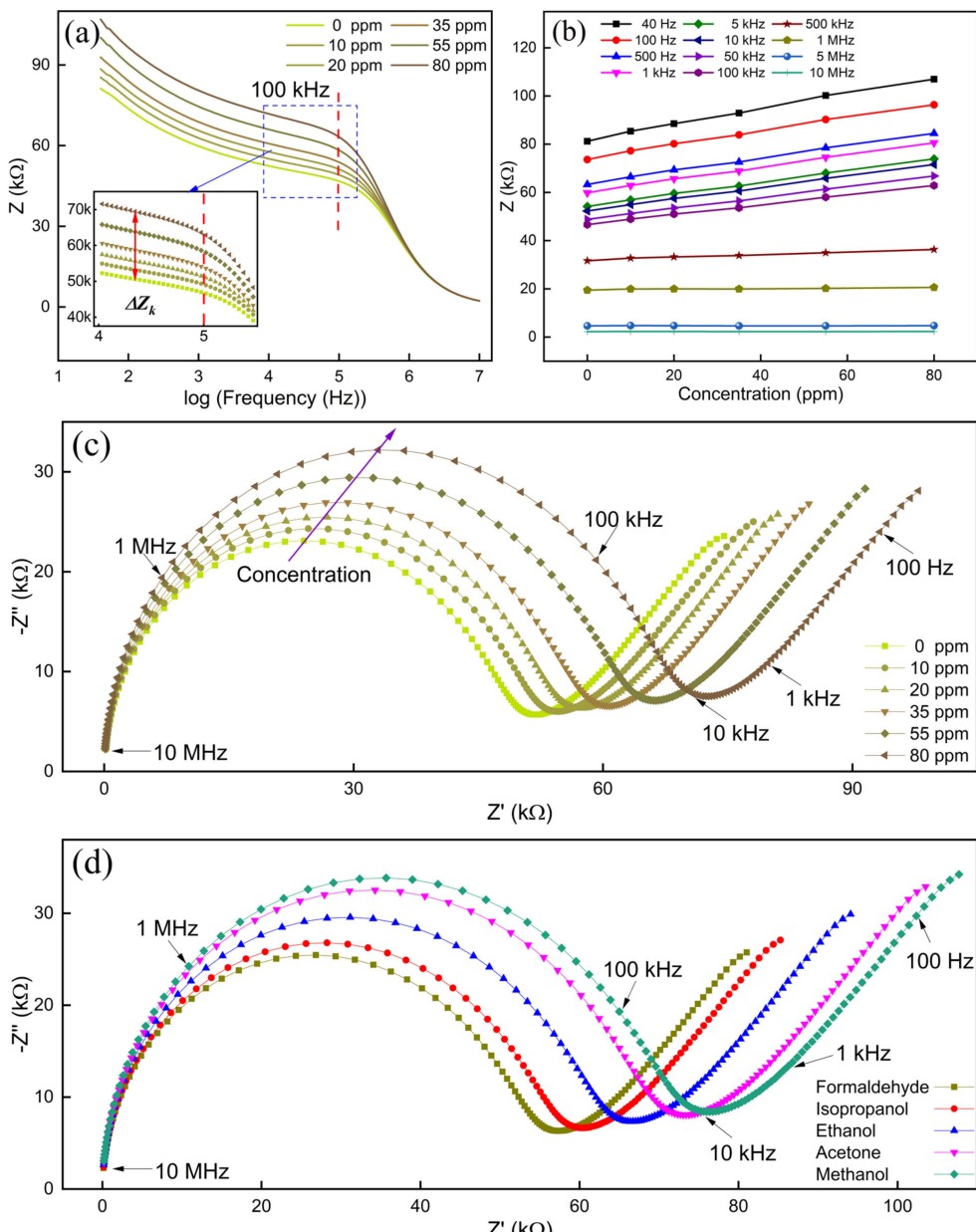

**Figure 5.** Impedance spectroscopy analysis of the sensor (eG, 180 °C, 1 L), (**a**) the Bode-plots under different formaldehyde concentrations; (**b**) calibration curves of the response impedance at extracted frequencies from (**a**); (**c**,**d**) the Nyquist-plots under different formaldehyde concentrations and different types of VOCs at 10 ppm, respectively.

Figure S2 shows the dynamic measurement of two alternating concentrations using $N_2$ flow carrying 80 ppm methanol and 0 ppm in. Comparison of measurement by DC and AC (100 kHz) is shown. Impedance measurement ($\Delta Z/Z_0$) shows faster response and recovery speed than that made by DC ($\Delta R/R_0$), which suggests that impedance can improve the response/recovery time significantly.

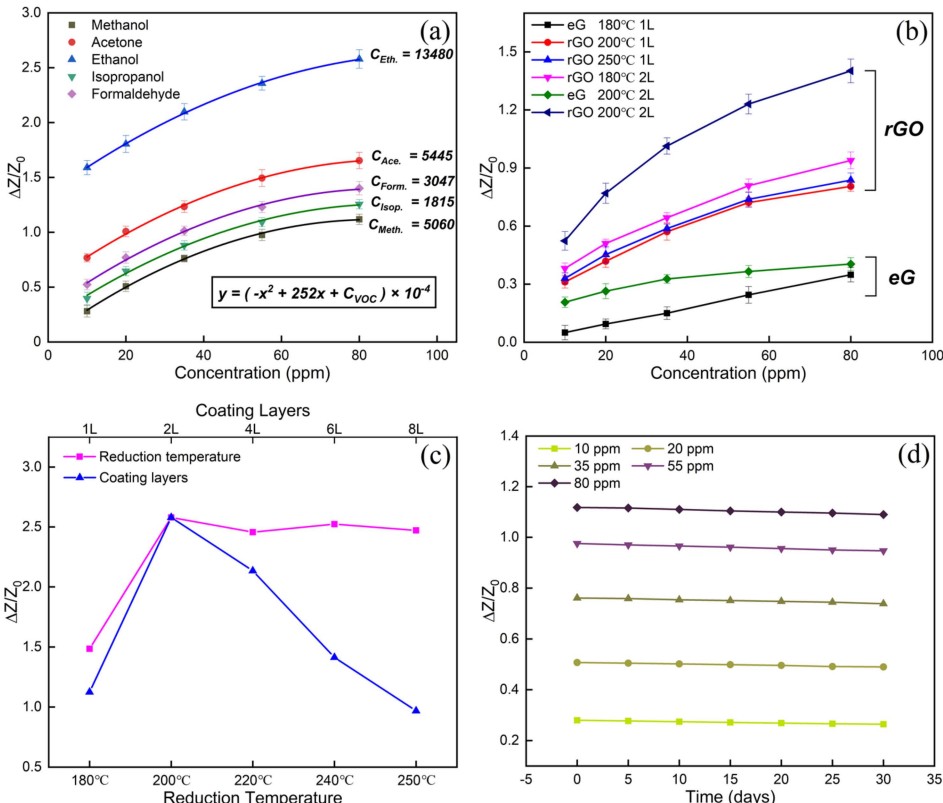

**Figure 6.** Sensing properties of the sensors constituting the multi-sensor array: (**a**) calibration curves of the sensor (rGO, 200 °C, 2 L) in dependence on the VOCs concentrations; (**b**) calibration curves of each sensor characteristic vs. formaldehyde concentration; (**c**) effects of the number of coating layers and of the reduction temperatures on the relative response change at 80 ppm ethanol; and (**d**) long-term stability of the sensor (rGO, 200 °C, 2 L) at a constant concentration of methanol (0–30 days).

Figure 6b shows the response curves for formaldehyde of the six sensors, finally selected to form an array. The rGO-based sensors generally have better response than graphene-based ones because defects remain in chemical GO after removing a large number of oxygen-containing functional groups; i.e., water and -COOH groups are removed (Figure 4), which enhances the ability of rGO to adsorb gas molecules [55]. The sensor curves exhibit different slopes, pointing to cross-validation based on an array for single VOC recognition and enhances qualitative accuracy [56].

The influence of RH in the measurement environment on the response of the proposed sensor was investigated. The RH of the experiment environment (inside the headspace) is shown in Figure S3, to be approximately 90%. The dynamic impedance measurement of the sensor (rGO, 200 °C, 2 L) towards an air flow of RH 90% is shown in Figure S4. The relative impedance change is 1.4%, which means that the RH in bottles has a low influence on VOC responses. The low response of well reduced GO towards humidity was reported in the literature [4,57].

Figure 6c shows how the number of coating layers and the reduction temperature affect the rGO response toward ethanol (80 ppm). The capacitance and electronic transport of graphene are known to vary with the number of layers, but its behavior as a function of the type of stacking layer is not clear yet.

The blue curve indicates that the relative change in the stacking layers of rGO (200 °C) thin-film casting sensors increases significantly as the coating layers increase to 2 layers (see Figure 6c). However, with further increasing of layers, it decreases strongly instead, due to the unavoidable thickness increase caused by the increasing coating layers number. The thickness affects the gas response. A relatively thin sensitive film leads to good sensitivity,

but the final effectiveness depends on various factors [58]. Electron transport in the whole sensor electrode probably decreases with layer number. Hence, charge accumulation due to VOC adsorption is reduced in the top rGO layer, making the less sensitive sensor.

The effect of thermal treatment under ambient room conditions was measured based on sensitive films of 2-Layers. The reduction temperature of 200 °C is relatively optimal in the range of 180 to 250 °C. For reduction temperature below 200 °C, only water vaporization and -OH groups removal are expected. However, above 200 °C, the reduction of the oxygen group can indeed stop occurring, giving place to re-oxidation of graphene, making it a turning point for the sensitivity of rGO and thus affecting the relative change in this temperature range [57,59].

The sensor's long-term (0–30 days) response stability (rGO, 200 °C, 2 L) toward methanol at different concentrations is demonstrated by monitoring the relative change toward methanol in Figure 6d. Sensor response decreases slightly with time. This phenomenon could be explained by gradual rGO oxidation in air and increase in the number of functional groups filling the graphene surface defects that can capture the gas molecules, reducing sensor responsivity. The higher the atmosphere concentration, the larger the corresponding decrease in relative change, from 111.8% to 109% on day 30 at 80 ppm. Due to the low-concentration response, the reduction cannot be well observed. However, the reliability is within the acceptable range, satisfying the demand for long-term stability in the EN application scenario [11].

### 3.4. Selectivity of Electronic Nose

The sensor array consists of six graphene-based sensors previously described in Table 1 and constitutes an EN system that recognizes VOCs. On the basis of VOCs "fingerprint" response analysis, an array can be constructed by non-specific sensors [12]. The greater the distinguishing ability, the more accurate the array response signal toward VOCs [11]. Figure 7 shows the selectivity of the sensor array. The sensors rGO-180-2L and rGO-200-2L have more intense response to VOCs.

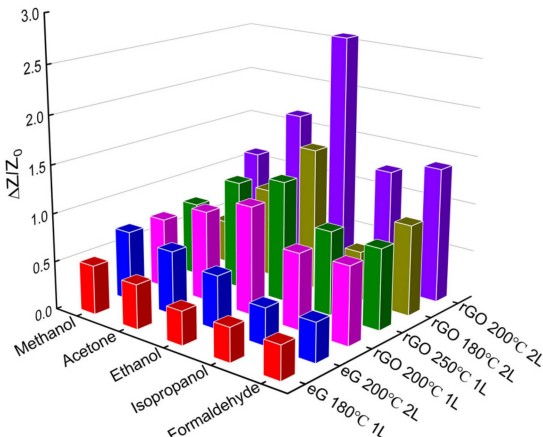

**Figure 7.** Comparison of responses from each sensor in the array toward the target VOCs at 80 ppm.

The eG sensors have less intense response than the rGO sensor, rich in surface defects. Such defects enhances the rGO sensor gas molecule capturing. For single sensors, rGO responses more strongly toward ethanol and acetone than other VOCs, whereas eG responses the best toward methanol and acetone. Most importantly, both rGO and eG shows unique response selectivity to all the measured VOCs, and each VOC elicited a unique response pattern. Using different thermal treatments for graphene-based materials is decisive for the selectivity of the sensor array and enhances the pattern recognition accuracy.

### 3.5. Analysis of Array Data

To evaluate the EN VOCs recognition performance, PCA was employed to analyze the response toward VOCs. The data were based on relative impedance change values of the sensor array described in Table 1. Figure 8a,b respectively show the score plots, loading plot, and scree plot of the response toward 10–80 ppm VOCs in 2D and 3D scales. Each arrow across every dot cluster indicates the increase in VOC concentration. For example, the blue dot on the left side of Figure 8a is obtained at the lowest ethanol. When the first two principal components (PC) are considered, the cumulative contribution reaches 95.5%, which means that the most information can be reasonably described [56]. Overall, the five VOCs can be distinguished. Regularly increase in the concentration placed on each cluster demonstrates excellent quantitative recognition potential. In terms of cluster distribution, methanol and acetone are distinguishable and completely independent of the other types, in agreement with other research [60].

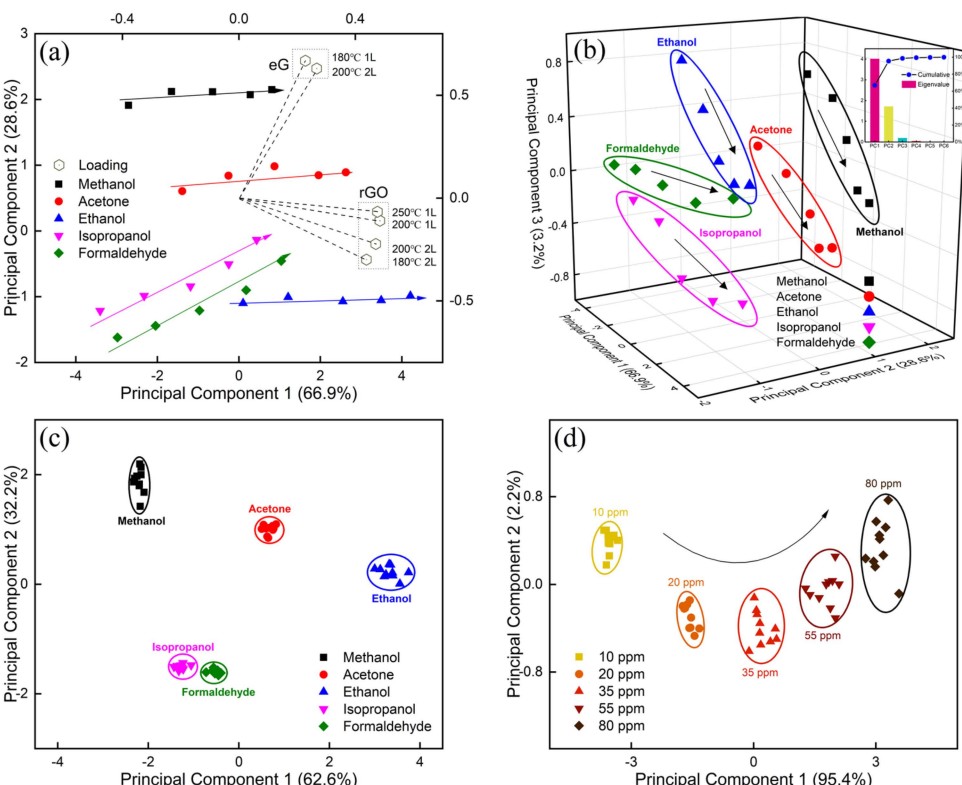

**Figure 8.** PCA results of the sensor array (two eG and four rGO sensors, as described in Table 1). (**a**) Score and loading plots toward target VOCs at different concentrations considering PC1 and PC2 (the upper and right axes indicate the loading results); (**b**) score plot toward target VOCs at different concentrations considering PC1, PC2 and PC3, inset shows the eigenvalues and cumulative contribution rate of each principal component in the scree plot; (**c**) qualitative recognition: score plot toward target VOCs at 20 ppm; and (**d**) quantitative recognition: score plot toward different isopropanol concentrations.

The isopropanol and formaldehyde distributions are parallel and sufficiently distinguishable, but with relatively narrow spacing. The low ethanol concentration overlaps little with the high formaldehyde concentration. Graphene-based materials present similar response towards ethanol, isopropanol, and formaldehyde [61]. Considering the third PC, the cumulative contribution reaches 98.7%. Because Z-axis projection is compressed, the discrimination of clusters in the 3D space is strengthened. The five clusters are completely separated and maintain linear concentration distributions. The isopropanol and formaldehyde distributions are no longer parallel but form a different angle in space and can be separated more precisely. The region low ethanol concentration has a large value

on the third PC (PC3) and has an observable distance from formaldehyde. The loading plot in Figure 6a (inset) shows how each sensor impacts the sensor array during the PCA calculation. The rGO and eG are partitioned into two tight clusters, indicating that the two materials play completely different VOCs recognition roles, but similar types of sensors play similar roles. The rGO and eG respectively contribute the highest on PC1 and PC2, so the rGO response property has a decisive effect on the contribution to sample space. At the same time, eG is crucial for the degree of dispersion in the array signal; i.e., the quantification of selectivity.

These findings are interesting because they suggest that the regions of graphene layers free of defects and/or functional groups contribute more to identify the adsorbed on the sensor (selectivity). Molecule adsorption onto graphene layers is expected to affect the graphene electronic structure, which in turn affects electronic transport and capacitance. In addition, defects are appropriate sites for trapping many molecules, but they are probably less sensitive to the molecule type. The rGO contains regions with and without defects, while the eG predominantly has a surface with a low concentration of defects.

The qualitative and quantitative recognition ability of sensor arrays is usually the crucial measure of performance. Figure 8c shows the PCA calculation results of the sensor array toward five VOCs at 20 ppm. The cumulative contribution of the two PCs reaches 94.8%, which is sufficient to describe the information of the array response. The clusters representing the VOCs are distributed completely independently in a 2D plane, so that the array can still achieve excellent qualitative recognition at low gas concentrations. The dots inside the clusters represent ten independent replicate measurements under the same environment, and the tightly aggregated distribution form shows the excellent repeatability of the sensors. The isopropanol and formaldehyde in Figure 8c clusters do not overlap, but they are closely arranged, which is consistent with Figure 8a. Such a problem can also be solved by adding an extra principal component such as PC3. Figure 8d demonstrates the PCA calculation results of the sensor array toward different isopropanol concentrations. The cumulative contribution of the two PCs reaches 97.6%. The clusters representing five concentrations are parabolically and completely independently arranged in a regular monotonic pattern with the concentration changes, meaning that recognition of concentration changes by the array is distinguishable and regular. Within the clusters, the independent measurement results are arranged in aggregates with good repeatability. Still, the area of the clusters increases slightly with increasing concentration, indicating that increasing concentration is positively correlated with increasing measurement error. Overall, the array demonstrated excellent qualitative and quantitative recognition ability toward VOCs.

### 3.6. Accuracy

The purpose of applying classifiers is to reduce the so-called "gray zone" by using classification algorithms to cut the areas of target modes precisely. SVM is favored in current machine learning due to its excellent and accurate hyperplane separation capability [62,63]. Ten independent measured data from the array towards five concentration atmospheres of each VOCs were utilized to train the PCA-SVM method, to achieve accurate VOCs recognition. Figure 9a shows the PCA-SVM classification results; there are 25 clusters representing VOCs categories and concentrations and 10 support vectors separating VOCs categories. The 2D plane is divided into five zones corresponding to the VOCs' categories by combining support vectors. Following the same trend as the previous PCA results, methanol and acetone can be completely and independently distinguished when only the first two PCs are considered. The low ethanol concentrations overlap slightly with the high formaldehyde and isopropanol concentration regions, which minimizes the formaldehyde zone. This weakness can be solved if PC3 is considered, as explained in Figure 6b. The information compressed in 2D projections (PC1—PC2) is released into the 3D space in the PC3 direction, avoiding intersection of the overlapping regions. The space blocks divided by plane vectors, representing VOCs categories, obtain clearer classification boundaries

than in 2D, significantly improving accuracy for the overlapping clusters, proven by the recognition results in Figure 9b.

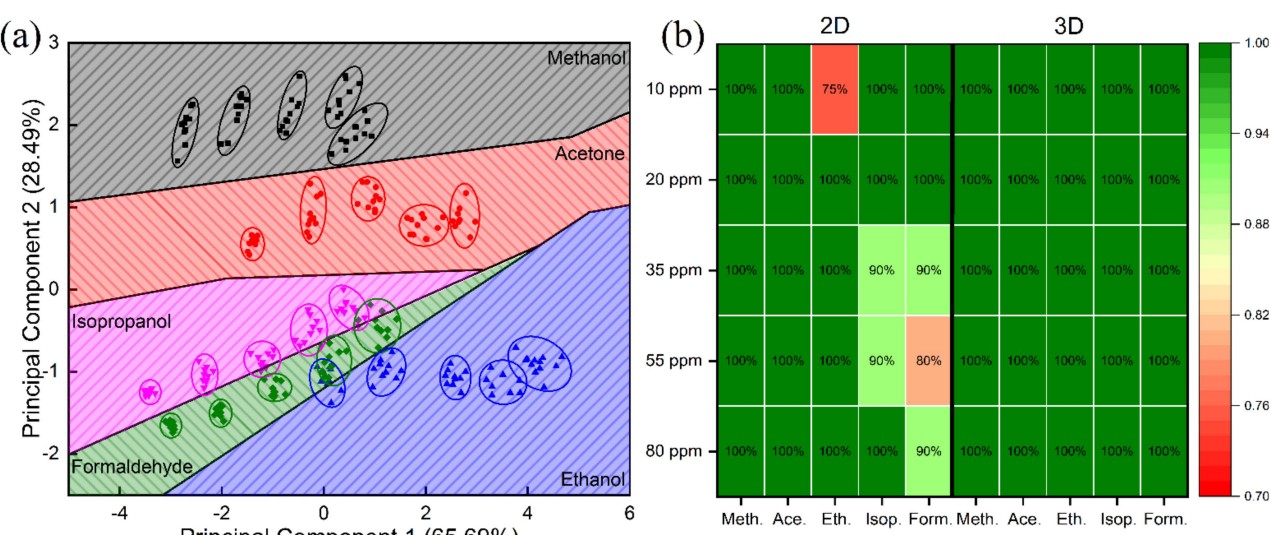

**Figure 9.** (**a**) Training result on the 2D-coodinate system with scores of PCA and support vectors of Linear-SVM towards target VOCs at different concentrations and (**b**) comparison of testing accuracy rate on qualitative and quantitative recognition under 2D- and 3D-coodinate system.

Figure 9b shows the accuracy of the develop system for qualitative and quantitative VOCs recognition in the 2D- and 3D-coordinate systems as a heat map. In the case of the recognition in the 2D-coordinate system, the array provides 75% accuracy at 10 ppm ethanol, 80–90% in 35–80 ppm formaldehyde and isopropanol, and 100% for the other conditions, following the same trend as the information analyzed in Figure 9a. Nevertheless, after considering PC3, overlapping clusters are eliminated, and the accuracy increases to 100%, which is better than when only PC1—PC2 are considered. The analysis results demonstrate the favorable potential of the sensor array in terms of recognition capability and robustness for both qualitative and quantitative recognition.

## 4. Conclusions

We have fabricated graphene-based IDE sensors with different responses towards VOCs by tuning the mechanical eG and chemical GO stacking layers and thermal reduction temperatures. We have built an impedimetric electronic nose for qualitative and quantitative VOC measurements. The EN consisted of two eG and four rGO sensors, two low and four highly defective graphene stacking layer sensors, respectively. The corresponding measurement system was applied under the guidance of impedance spectroscopy analyses. The EN has high relative response change, 257% in 80 ppm ethanol when the sensor (GO, 200 °C, 2 layers) is employed towards five human health-related VOCs; in addition, it exhibits excellent selectivity and long-term stability. It also demonstrates an acceptable response under low concentration conditions of 10 ppm with a minimum relative change of 5% (formaldehyde; eG, 180 °C, 1 layer). PCA results demonstrate that the EN recognizes VOCs categories and concentrations when considering three principal components. The signal processing based on PCA and SVM shows a high accuracy of the e-nose. The potential for more VOC categories is achievable, and a wider concentration range is detectable. We envision that the developed EN is suitable in many applications, such as indoor air quality analysis and early diagnosis in health care (e.g., cancer). On the other hand, to make EN feasible, some challenges still should be overcome, including ppb-level VOCs detection and recognition of complex mixtures. To solve these challenges, we plan to increase the utilization of the structural properties of graphene and its hybridization in combination

with advanced machine learning algorithms to realize further improvements in sensitivity, selectivity, and accuracy of the EN.

**Supplementary Materials:** The following are available online at https://www.mdpi.com/article/10.3390/chemosensors9120360/s1, Figure S1: The dynamic impedance response of the rGO sensor in reference and VOC bottles al-ternatively, Figure S2: The dynamic response of the rGO sensor under DC and AC measure-ment conditions, Figure S3: The real RH of VOC bottle measured by commercial reference sen-sor SHT85, Figure S4: The influence of RH 90% on the relative impedance change of rGO sensor.

**Author Contributions:** Conceptualization, A.A.-H. and O.K.; methodology, T.L. and A.A.-H.; soft-ware, T.L., J.H. and Y.W.; validation, T.L., Z.H. and T.Y.; formal analysis, T.L.; investigation, A.A.-H., J.M.R. and E.Y.M.; resources, O.K.; data curation, A.A.-H. and A.A.; writing—original draft prepa-ration, T.L., A.A.-H. and Z.H.; essay—review and editing, T.L., A.A.-H., J.M.R., Z.H. and O.K.; visualization, T.L. and J.H.; supervision, O.K.; project administration, O.K. and A.A.-H.; fund-ing acquisition, O.K. and A.A.-H. All authors have read and agreed to the published version of the manuscript.

**Funding:** T.-Q.L., A.A.-H. and O.K. acknowledge the financial support within the project Nutricon (No. 24119201) funded by the Sächsische Aufbaubank (SAB). O.K. and A.A.-H. acknowledge the financial support Deutsche Forschungsgemeinschaft (DFG) (PhotoSens project no. KA 1663/12-1).

**Acknowledgments:** The authors thank Yang Pan and Dietrich R.T. Zahn, the professorship of Semiconductor Physics, Chemnitz University of Technology, Chemnitz, Germany, for access to the Raman spectroscopy device and Leonardo Paterno, Instituto de Química, Universidade de Brasília, Brasilia, Brazil, for the fruitful discussion about electronic nose and PCA methods.

**Conflicts of Interest:** The authors declare no conflict of interest.

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
