# Peer review of "Flexible Impedimetric Electronic Nose for High-Accurate Determination of Individual Volatile Organic Compounds by Tuning the Graphene Sensitive Properties"

_chemosensors, doi:10.3390/chemosensors9120360_

Round 1
Reviewer 1 Report
This manuscript demonstrates functionalized graphene materials for the detection of VOCs, such as formaldehyde, methanol, ethanol, acetone, and isopropanol. The design of the VOCs gas sensor was attractive and complete. The manuscript can be considered for publication after the minor comments.
- In Figure 1, the designed silver IDE sensor was placed in a bottle with a liquid mixture. The description of the sensor system was absent, such as the size of the bottle.
- In section 3, a series of experiments was performed to acquire the accuracy of the electronic nose. The concentration of the different VOCs gas was obtained by Henry's Law. According to the Conclusions, “an acceptable 527 response under low concentration conditions of 10 ppm with a minimum relative change 528 of 5% (formaldehyde, eG-180-1L).” The released gas from the liquid mixture was imprecise. For a better sensor performance evaluation, the authors should use a standard gas cylinder and put the developed IDE immersed in the standard gas.
Reviewer 2 Report
The authors present a chemical sensor to measure VOCs using graphene/graphene oxides and impedance measurement. The sensor array composed of two eG and four GO sensors has successfully demonstrated detecting multiple VOCs. However, this reviewer considers the following questions to be addressed for more clarification of the manuscript;
1) In the introduction, the use of impedance measurement can promote faster response/recovery time. However, the manuscript does not discuss the response/recovery time to support the benefit of using an alternating current.
2) The samples appear to be dried under a nitrogen flow. What about the environment for thermal heating? Is it also under nitrogen flow?
3) Target VOCs are dissolved in water for the sensor measurement. Since the humidity is reported to influence the sensing signals, the authors need to provide the humidity value during measurement. In addition, the influence of relative humidity levels on gas sensing properties should be discussed.
